# The nexus between the attitudes and self-concepts of gifted students in an Arab context

**Maxwell Peprah Opoku**[1]*, **Ashraf Moustafa**[1], **Negmeldin Alsheikh**[2], **Noora Anwahi**[1], **Mariam Aljaberi**[1], **Thara Alkhateri**[1], **Aysha Almeqbaali**[1], **Hala Elhoweris**[1]

1 Department of Special and Gifted Education, United Arab Emirates University, Al Ain, United Arab Emirates, 2 Curriculum & Methods of Instruction, United Arab Emirates University, Al Ain, United Arab Emirates

* Maxwell.p@uaeu.ac.ae

## Abstract

### Background

The research literature has reported the need for educators to develop suitable programs geared toward the nurturance of gifted students. Although some non-Western countries have adopted policies to foster the growth of exceptional students, their contributions to research in gifted education are limited. To expand the literature on gifted education, this study was guided by Ajzen's theory of planned behavior to explore the attitudes and self-conceptions of gifted and talented students in the United Arab Emirates.

### Methods

The revised Self-Perception Profile for Children and Opinions About the Gifted and Their Education scales were used to collect data from 150 high school students in Abu Dhabi, one of seven states (emirates) in the United Arab Emirates. AMOS software version 29 was used to conduct confirmatory factor and path analyses. To evaluate the differences between the background variables on attitudes and self-concepts, t-tests were calculated using SPSS software version 28.

### Results

The study found a relationship between the attitudes and self-concepts of students regarding their participation in enrichment programs. The hypothesized moderation effect of demographic variables on the relationship between attitudes and self-concepts was not supported.

### Conclusion

The study's limitations and implications for educators in the United Arab Emirates to prioritize programs geared toward developing the attitudes and self-concepts of gifted students are discussed in detail.

**Data Availability Statement:** Data cannot be shared publicly because of students' privacy as those on the high ability list are publicly known. Data are available from the Social Science Ethics

Committee at United Arab Emirates University
(research.office@uaeu.ac.ae) for researchers who
meet the criteria for access to confidential data.

**Funding:** Office of the Associate Provost for
Research at United Arab Emirates University. The
funders had no role in study design, data collection
and analysis, decision to publish, or preparation of
the manuscript.

**Competing interests:** The authors declare that they
have no competing interests.

## Introduction

The need to develop suitable programs for gifted and talented (GT) students has been well explored [1, 2]. There is a consensus that GT students can become highly productive citizens who contribute to enriching the communities in which they live by finding solutions to social problems [1, 3]. It has been suggested that GT students should be enrolled in appropriate programs that encourage their potential and ability to engage in social problem solving [2, 3]. To ensure that GT students derive the maximum benefit from such programs, researchers have paid attention to their self-concepts [4–6] and attitudes toward participating in them [7–10]. Such studies, however, have a relatively limited scope because most have been undertaken in highly developed countries, such as the United States [11, 12]. Since GT programs are potentially influenced by the culture and way of life in a given context [13–15], this study aimed to explore the relationship between the attitudes and self-concepts of GT students in a novel Arabian context, the United Arab Emirates (UAE).

The GT concept is not without controversy, and there are two main worldviews regarding giftedness. The first school of thought associates giftedness with individuals who excel academically and score at least 140 on IQ testing. This concept of giftedness has been criticized as narrow, restrictive, and favoring a particular group of students from privileged backgrounds [14–16]. Stack et al. [16] argued that defining giftedness according to cognitive ability limits or excludes the enrollment of students from diverse or minority cultural backgrounds in gifted programs. They pointed out that students whose first language is not English are at a disadvantage in enrolling in such programs because the instruments and measurements used to identify GT students are primarily in English. The second worldview is more culture-based and argues that GT is not only related to cognitive ability but also to individuals' ability to manage human relationships [16–18]. Consequently, there have been calls for the inclusion of informal processes in the concept of GT [13–16]. Even though the GT lens has a limited scope in the UAE and other comparable cultural contexts, there is an ongoing quest for best practices and strategies to promote the development of GT students in these countries [19].

The placement of GT students in programs has been described as labeling [5, 11], and there are diverse views concerning ability groupings. For example, one school of thought contends that the identification and subsequent placement of GT students in gifted programs builds their confidence and elicits their best, resulting in improved behavior [11]. The advantages of ability groupings have spurred educators to recognize and place GT students in programs that will boost their development. Conversely, researchers have contended that ability grouping such as enrichment programs is counterproductive as it drives a wedge between those identified as GT and others [5]. As a result, it is challenging for GT students and their peers to form healthy relationships in the same classroom. In addition, some GT students might be complacent because they feel advantaged and unchallenged to work hard to reach their potential [1]. However, despite the risks of identifying and grouping GT students, educators and policymakers should develop initiatives that support them to maximize their potential. GT students will benefit the most from placement in programs that allow their gifts to unfold and fully develop.

Western countries have more experience in developing gifted programs compared with non-Western countries. While the UAE has developed gifted programs for GT students [19], no attempt has been made to understand the attitudes and self-concepts of students enrolled in such programs. The overarching aim of the current study was to investigate the relationship between the attitudes and self-concepts of GT students toward their inclusion in gifted programs.

## Theoretical framework

This study was guided by Ajzen's [20–22] theory of planned behavior (TPB), a major theory used to assess human behavior in societies [23]. The TPB expands on the theory of reasoned action, arguing that behavior is a product or interplay of two related beliefs, behavioral and normative, which impact the intention toward a given behavior [20, 24]. Normative beliefs, in contrast to behavioral beliefs, refer to the effect or support of important others on one's intention to perform an action. Behavioral beliefs pertain to one's view of a certain behavior [22]. However, Ajzen [20] later expanded his theory to include a third variable, control beliefs, which might directly impact one's actual behavior. Control beliefs refer to a person's confidence in their capacity to complete a given task or behavior [20, 21]. The three related beliefs combine to predict one's intention toward a given behavior. However, although the TPB has been used to guide studies on inclusive education for students with disabilities [22, 23], it has rarely been applied to studies of GT students.

For the purpose of this study, the tenets of TPB were adopted. According to Ajzen [20], these tenets develop into variables that predict human behavior. For example, behavioral beliefs develop into attitudes toward a given behavior, normative beliefs develop into subjective norms, and control beliefs develop into perceived behavioral control (used as a proxy for self-concept in this study). In this study, attitudes were operationalized as GT students' perceptions of being categorized and participating in gifted programs. Additionally, subjective norms were explained as the effect or external influence on GT students' involvement in gifted programs. Moreover, self-concept is defined as GT students' understanding of their personalities and confidence in their abilities. According to expectancy-value logic, favorable attitudes toward the three intention components are critical for achieving a desirable outcome [21, 22]. Specifically, favorable attitudes, subjective norms, and positive self-concepts are vital to GT students' success. In other studies on teachers' attitudes toward students with disabilities using the TPB, researchers have reported that favorable attitudes and self-concepts contribute to effective classroom practices [25, 26].

Ajzen [20, 21] further suggested that the cumulation of subjective norms, attitudes, and self-efficacy explains one's intention to perform a given behavior. The focus of this study, however, was on people's attitudes and self-concepts, because they have direct control over these. Ajzen [20] suggested a relationship between students' attitudes and self-concepts and reasoned that they need to possess favorable attitudes and self-concepts before engaging in a given behavior. In this study, a relationship between attitude and self-concept was proposed. This hypothesis has been tested and supported in studies on inclusive education [25, 26].

Ajzen [21] also suggested that the background variables of individuals involved in a given behavior could further explain their intentions. Therefore, human behavior researchers should pay close attention to demographic characteristics and their effects on individual actions or inactions. In this study, it was predicted that background variables could influence the link between attitude and self-concept. In inclusive education research, the TPB has been used to study the association between demographic variables and intentions toward teaching practices [25, 26]. However, these studies have yielded contradictory results, and contextual studies of intention are therefore required to develop insights and influence teaching practices. Against this backdrop, in this study, TPB was used as a lens to study the relationship between the attitudes and self-concepts of GT students in the UAE.

## Attitudes of gifted students

In this study, attitudes refer to students' perceptions of being in a gifted program. According to Gagné [27], understanding attitudes toward gifted education is imperative to develop

baseline information about the effectiveness of practices in schools. However, studies on attitudes toward gifted education have been limited mainly to preservice [28] or in-service teachers [12, 29]. A large study in Ireland examined teachers' attitudes toward teaching GT students. Although the teachers indicated that they supported gifted education and used specialists to support the teaching of GT students, they had an unfavorable attitude toward grade acceleration. Similarly, in a Swedish study conducted by Ivarsson [30], principals rated their attitudes using the Opinions About the Gifted and Their Education (OGE) scale developed by Gagné and Nadeau [31]. The principals' perceptions of gifted education ranged from negative to ambivalent. However, it is apparent that GT students have not been given many opportunities to share their attitudes toward being in a gifted program.

There is a general consensus that children are "rights bearers" who have the capacity to share or contribute to issues concerning their well-being. Unfortunately, studies on the attitudes of students, such as those in secondary schools, toward their participation in gifted programs are very few. Only a small body of literature has looked at the attitudes of GT students [32] toward gifted programs such as environmental [9, 10] and science programs [7, 8]. In Turkey, Özarslan [9] conducted studies of GT students participating in a gifted program organized by the Science and Art Center (Bilsem). Both surveys revealed comparable results, demonstrating students' overall positive attitudes toward plants and recycling. However, in the latter study, there was no discernible difference between GT students in terms of age, gender, grade, or years of attendance. In Singapore, studies by Caleon [7] and Lang [8] showed a relationship between intellectual ability and students' attitudes toward science. Caleon [7] reported that when asked about pursuing science in the future, students with above-average academic abilities and male students were more inclined to respond positively than other groups. Lang [8] discovered that gifted female students had more favorable attitudes toward their interactions with chemistry teachers than male students. Although these findings provide valuable insight into GT students' attitudes toward specific subjects, there are no comparable findings from the UAE.

### Previous studies on GT students' self-concepts

Self-concept has been defined as students' confidence in their abilities and potential. There is a link between student well-being and academic performance [33, 34], and underlying this relationship is the self-concept. Researchers have suggested that students with high academic achievement possess positive self-concepts [35]. This finding has highlighted the need for education systems to develop positive self-concepts among students. Consequently, many studies have compared the self-concepts of GT and non-GT students [11, 12, 30, 35–37]. These studies have yielded contradictory findings. Whereas some studies have reported no difference in self-concepts between gifted and non-GT studies [33, 36], other studies have shown different results [12, 37]. For instance, in a study conducted across 28 school districts in the United States, McCoach and Rinn [12] indicated a variance between the self-concepts of underachieving and high-achieving students. They reported that GT students scored higher on goal valuation and self-regulation compared to their underachieving counterparts. However, one area of consensus in the research is that gifted and non-gifted students differ in their academic self-concepts [33]. Therefore, to expand the discussion on the importance of self-concept in academic performance, more comparative studies between GT students and non-GT students are needed to inform educational policy and practice. However, in the UAE, there is no information about the self-concepts of GT students compared with those of their non-gifted counterparts.

Researchers have paid attention to studying the self-concepts of GT students [32]. Many studies have reported differences between students based on their self-concept [4–6, 12, 33, 38, 39]. For instance, a review paper by Litster and Roberts [33] compared the differences between

GT and non-GT students. On athletics and perceived looks, GT students reportedly scored lower than non-GT students. Gifted students, however, outperformed non-gifted students in both academic performance and perceived behavior. Since the UAE is currently developing its system to encourage gifted education [19], more scholarly attention needs to be paid specifically to students involved in gifted programs beyond comparison studies with students who are not involved in gifted programs.

## Overview of the study

Following the 2008 educational policy in the UAE, the Ministry of Education formulated the country's educational strategy goal of 2010–2020, which was a student-centered approach to education [19, 40]. The strategic plan emphasized that every student with an exceptional learning need requires individualized education to enable them to achieve their learning goals. Additionally, the Ministry of Education launched the flagship gifted education enrichment program entitled the "Development of Gifted and Talented Students' Skills." During the 2014 calendar year, the program underwent restructuring and was rebranded the "Integrated System to Identify and Care for Talents." Schools were expected to encourage high achievers to investigate social problems and use innovative problem-solving approaches to solve them [40].

These policy initiatives have contributed to the formation of interest groups and associations supporting the education of GT students across the UAE. Notable is the Hamdan Foundation, which promotes gifted education, teacher development, public awareness, parental training, and the implementation of gifted programs in the UAE [19, 40]. Although empirical evidence has revealed discrepancies and inadequate enactment of gifted education in the UAE [19], there is a dearth of empirical studies investigating students' learning experiences in these programs.

## Current study

Based on Ajzen's [20] conception of the TPB, a proposition was made in this study with respect to the relationship between self-concepts and attitudes. According to Gagné [27], the extent to which aspects of attitude predict other variables, such as self-concept, has not received much scholarly attention. In view of this, we hypothesized the following:

**Hypothesis I**: GT students' self-concepts will predict their attitudes toward their involvement in gifted programs (see Fig 1).

Additionally, demographic variables have been identified as vital in efforts to understand the intentions toward a given behavior. For example, in the UAE, gender is a major factor in the grouping of secondary schools according to their high-ability areas, and, typically, secondary school students attend single-sex classrooms. Consequently, their enrollment in gifted programs has followed a gendered pattern. In this study, gender and high-ability area were selected as variables of interest to explore their impact on the relationship between students' self-concepts and their attitudes toward gifted programs:

**Hypothesis II:** Demographic variables will be a significant moderator of the attitudes and self-concepts of gifted students.

The available evidence has indicated that GT students' attitudes and self-concepts have been studied individually. However, there is little or no empirical research on the relationship between GT students' attitudes and self-concepts. Studies examining students' attitudes and self-concepts are also sporadic in Arabian contexts. Therefore, there is a need to examine the

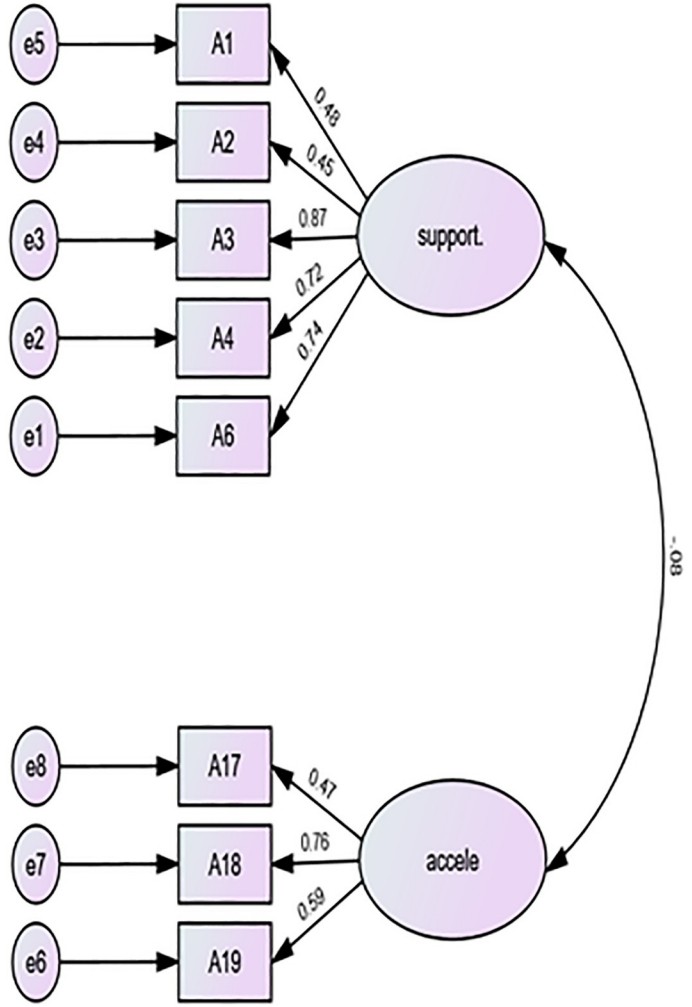

**Fig 1. Confirmatory factor analysis for an opinion about the gifted and their education.**

relationship between attitudes and self-perceptions among GT students in the UAE. This study aimed to answer the following research questions:

1. What are the differences between GT students in demographic variables, attitudes, and self-concepts?

2. Do GT students' self-concepts predict their attitudes toward participating in gifted programs in the UAE?

3. Do demographic variables moderate the self-concepts and attitudes of GT students toward participating in gifted programs in the UAE?

## Methods

### Design

The study entailed cross-sectional, descriptive research that aimed to capture the self-concepts of students at a given time [41]. The design was appropriate as it enabled the study of students' attitudes and self-concepts at a given point in time.

The participants in this study were GT high school students nominated by their schools to participate in an enrichment program. In the UAE, there is a three-tier educational structure: cycle one (pre-school to grade 4), cycle two (grades 5–8), and cycle three (grades 9–12). While public schools are managed by the Emirates Schools Establishment, they are also supervised by the Abu Dhabi Department of Education and Knowledge (ADEK). In this study, GT students were identified based on their performance on IQ tests conducted in the schools. GT students participate in projects related to the science, technology, engineering and mathematics (STEM) areas of high ability. For instance, some of the students have been involved in the Maker's Real Engagement in Active Problem-Solving project [42] on environmental pollution.

Simple random sampling was used to recruit the study participants from the sampling frame [43]. Each student involved in the enrichment program was invited to participate in this study. The current study drew GT students from private schools, as they were the main schools running enrichment programs for GT students. The decision to include students from private schools was based on ADEK granting its permission to include students on their high-ability roster. Prior consent was sought from all parents whose children were on the ADEK register. Invitations were sent via email to all GT students participating in the enrichment program. A total of 150 students took part (see Table 1).

## Data collection

This study forms part of a larger project that sought to understand the well-being of GT students participating in an enrichment program in the UAE [44]. The study was approved by the Social Sciences Ethics Review Committee of the United Arab Emirates University (ERS_2021_8436).

**Table 1. Difference between background variables, attitudes, and self-concept.**

|  | Sample | Attitude | Self-concept |
|---|---|---|---|
| **Gender** |  |  |  |
| Male | 37 (25%) | 3.62 (.54) | 3.77 (.65) |
| Female | 113 (75%) | 3.66 (.55) | 3.66 (.61) |
| *t* |  | -.34 | .91 |
| *Cohen's d* |  | .06 | .17 |
| **Age** |  |  |  |
| 10–15 years | 38 (25%) | 3.62 (.56) | 3.66 (.57) |
| 16–18 years | 112 (75%) | 3.66 (.54) | 3.69 (.64) |
| *t* |  | -.33 | -.27 |
| *Cohen's d* |  | .06 | .05 |
| **Grade Level** |  |  |  |
| Intermediary level (Grades 7–9) | 40 (27%) | 3.65 (.65) | 3.75 (.66) |
| High School (Grades 10–12) | 108 (73%) | 3.65 (.51) | 3.66 (.61) |
| *t* |  | .009 | .08 |
| *Cohen's d* |  | .002 | .15 |
| **Area** |  |  |  |
| STEM areas | 110 (73%) | 3.64 (.55) | 3.69 (.66) |
| Other areas | 40 (27%) | 3.68 (.54) | 3.67 (.51) |
| *t* |  | -.37 | .15 |
| *Cohen's d* |  | .07 | .03 |

STEM: science, technology, engineering and mathematics

The survey was made up of three sections. The first section collected demographic data (gender, age, grade level, and high-ability area(s)), which were included based on a review of the previous literature [8–10, 12].

The ensuing parts of the instrument were published scales related to the components (attitudes and self-concepts used as a proxy for self-efficacy) of Ajzen's [20] TPB, which was adapted for this study. Specifically, the second part [45] used a revised Self-Perception Profile for Children (SPPC) scale, which has been used extensively to study the self-concept.

The scale is made up of 34 items scored on a 5-point scale: strongly disagree (1) to strongly agree (5). The scale comprises six subscales (scholastic competence [students' rating of their academic ability], $n = 5$; social acceptance [ability of students to develop healthy social relationships], $n = 6$; athletic competence [competence of students in extracurricular sporting activities], $n = 6$; physical appearance [extent to which students accept themselves], $n = 5$; behavioral conduct [students' ratings of their emotional regulations], $n = 6$; and global self-worth [defined as how students appreciate and embrace themselves], $n = 6$).

The third part was a revised 20-item OGE scale [31]. The OGE scale was adopted for this study because it has been used extensively to study teachers' attitudes [27, 32]. However, in Gagné's [27] view, the instrument needed to be more efficient and concise to capture the core components of attitudes. Based on feedback from expert reviewers, the authors settled on the revised instrument, which is short and captures the important aspects of attitude that can be reported by secondary school students. Additionally, this appears to be the first time the OGE has been implemented for students in a non-Western context; thus, we needed to use a version that students could easily comprehend. These justifications informed the use of the abridged version of the original instrument. Responses to the measure, which consists of three subscales (support, acceleration, and elitism), are anchored on a 5-point Likert scale ranging from strongly disagree (1) to strongly agree (5).

The instrument consisted of three subscales. First, support for gifted education is comprised of 10 items and refers to the extent to which participants agree with their involvement in a special program for the gifted. Elitism, which comprises six items, assesses participants' beliefs as to whether being involved in a gifted program represents favorable or preferential treatment given to gifted students. Acceleration, constituting four items, measures beliefs about the acceleration of gifted students in schools [12].

Three specialists were given the questionnaire to assess and remark on its suitability for data gathering. This process is called the Delphi approach [46], which is a review of instruments by experts with in-depth knowledge who evaluate the suitability of a questionnaire before its implementation. Two versions of the instrument were given to the experts so that they could determine the version most suitable for data collection. There was consensus among the specialists on the use of the short version for data collection for this study. Following this, the instrument was piloted with eight GT students who were not considered for participation to determine whether it was clear and simple to understand.

An official invitation was sent to ADEK seeking permission to conduct this study. The parents of children on the high-ability roster gave ADEK their consent before their email contacts were released to the research team. The parents were informed of the study's goals and provided a breakdown of its methodology. They were also assured that no identifiable information about their children would be used or shared with anyone outside the research team. The parents were further informed that their children would not be incentivized to participate in this study. The children whose parents responded positively provided informed consent for participation in this study.

Google Forms was employed for data collection due to the mobility challenges imposed by the COVID-19 pandemic. There were Arabic and English translations of the items on the

instrument. The data were collected from May 2022 to July 2022. The duration of the survey completion was 15–25 minutes.

## Data analysis

The data were cleaned in Microsoft Excel before being imported to the Statistics Package for Social Science (SPSS) version 28 for analysis. The initial computation of missing data using missing at-random tests showed a range of 0.6% to 1.7%. Following this, the missing data were imputed. After that, a normality test was performed to determine whether the data were appropriate for parametric analysis. The results of the Kolmogorov–Smirnov and Shapiro–Wilk tests were insignificant, suggesting the normality of the data for parametric testing [47].

While the SPPC was validated in an earlier study [44], the OGE was used for the first time in the current context and thus, tt was necessary to assess its construct validity. Consequently, confirmatory factor analysis (CFA) was computed to understand OGE's dimensionality. The following goodness-of-fit indices were used: a chi-square value of less than 0.5, a comparative fit index (CFI) and a Tucker-Lewis index (TLI) of 0.90 or greater, a root square error of approximation (RMSEA), a standard root mean square residual (SRMR) of at least 0.08, and a regression weight of at least 0.50 [48, 49]. However, if the model did not meet the above thresholds, modification indices were observed to eliminate items with high and erroneous covariances [50].

The computation CFA for the 20-item OGE scale showed a poorly fit model: chi-square = 3.02, CFI = 0.60, TLI = 0.53, RMSEA = .012, and SRMR = 0.11. The items on the elitism subscale had poor loadings (items 11, 12, 13, 14, 15, and 16), and thus were deleted from the model. Additionally, half of the items of the support subscale (5, 7, 8, 9, and 10) had factor loadings below the cutoff and were removed from the model (see Fig 1). The rerun reached acceptable indices: chi-square = 1.12, CFI = 0.99, TLI = 0.98, RMSEA = 0.03, and SRMR = 0.05. However, there was no correlation between the subscales support and acceleration ($r = -.08$).

The instrument's reliability was assessed using Cronbach's alpha: attitude (0.60) and self-concept (0.90). The outcomes confirmed that the tool was suitable for reporting the study.

For Research Question 1, which concerned the effects of GT students' demographics on attitudes and self-concepts, t-tests were used because all the demographics had two levels. An assumption of variance homogeneity was observed to ensure that it was not violated [47].

To answer Research Question 2, direct path analysis, a form of structural equation modeling, was used to test the relationship between attitudes and self-concepts. Path analysis was used because it is appropriate for testing hypotheses [50], such as the assumed relationship between attitude and self-concept.

A preliminary CFA was used to determine the relationship between overall attitudes and self-perceptions, and the results were interpreted using the following thresholds: small (0.10–0.30), moderate (0.31–0.50), and large ($\geq 0.51$) [47]. The contribution of self-concept to the variance in attitudes was computed using path analysis. Based on Gagné's [27] suggestion that researchers investigate the types of attitudes and the specific behaviors they influence, path analysis was conducted at the subscale levels. To ensure that the model was appropriate, the fit indices stated above were observed.

For Research Question 3, the two demographic variables of interest—gender and students' ability areas—were added to the earlier path analysis model to assess their interaction effects on the association between attitudes and self-concepts. Both bootstrap and bias-corrected confidence intervals were set at 500 and 95%, respectively. In cases where there was a significant

moderation effect, Andrew Hayes's [51] PROCESS model was used to explore the direct effect of the moderators on attitudes and self-concepts.

## Results

### Association between demographics, attitude, and self-concept

The computation of mean scores showed students' neutrality on attitudes ($M = 3.65$; $SD = 0.54$) and self-concepts ($M = 3.69$; $SD = 0.62$). Additionally, the mean scores for self-concept showed that participants were high on global self-worth ($M = 4.12$; $SD = 0.91$) but ambivalent on the remaining subscales (scholastic competence, $M = 3.94$, $SD = 0.74$; social acceptance, $M = 3.49$, $SD = 1.20$; athletic competence, $M = 3.01$, $SD = 1.11$; physical appearance, $M = 3.42$, $SD = 1.29$; behavioral conduct, $M = 3.93$, $SD = 1.29$; see Table 2).

Concerning the subscale on attitudes, the participants were ambivalent about support ($M = 3.71$, $SD = 1.02$) and acceleration ($M = 3.56$, $SD = 1.10$; see Table 3).

To determine whether there was an association between background variables, attitudes, and self-concepts, independent t-tests were conducted. The outcomes revealed no differences in the participants' attitudes or self-perceptions (see Table 1).

**Table 2. Summary of mean scores on self-concept.**

|  | Items | M | SD |
|---|---|---|---|
|  | **Scholastic competence** | 3.94 | .91 |
| W1 | I am good at schoolwork | 4.34 | .82 |
| W3 | I remember lesson easily | 3.77 | .94 |
| W4 | I perform very well in science and mathematics | 3.92 | .94 |
| W5 | I figure out answer easily | 3.72 | .93 |
|  | **Social acceptance** | 3.49 | 1.20 |
| W7 | I have a lot of friends | 3.71 | 1.24 |
| W9 | I do things with a lot of friends | 3.64 | 1.15 |
| W11 | I am very popular in school | 3.12 | 1.21 |
|  | **Athletic competence** | 3.01 | 1.11 |
| W12 | I am very good at sports | 3.32 | 1.10 |
| W15 | I am better than others in sports | 2.79 | 1.15 |
| W17 | I am good at both outdoor and indoor sports | 2.92 | 1.07 |
|  | **Physical appearance** | 3.42 | 1.29 |
| W18 | I am very happy with my physical appearance | 3.48 | 1.22 |
| W19 | I am happy with my height and weight | 3.31 | 1.34 |
| W20 | I like my body as it is | 3.46 | 1.27 |
| W21 | I like my physical appearance as it is | 3.42 | 1.32 |
|  | **Behavior conduct** | 3.93 | 1.29 |
| W23 | I like my behaviour | 4.13 | .94 |
| W24 | I usually do the right thing | 3.77 | 1.00 |
| W25 | I usually do things in the right way | 3.83 | .89 |
| W28 | I behave well all things | 3.98 | .97 |
|  | **Global self-worth** | 4.12 | .97 |
| W29 | I am happy with myself | 4.12 | .96 |
| W31 | I am happy with myself as a person | 4.12 | .95 |
| W32 | I like myself | 4.17 | .97 |
| W33 | I am happy the way I am | 4.05 | 1.01 |

**Table 3. Summary of mean scores on attitude towards gifted education.**

|     | Item | M | SD |
| --- | --- | --- | --- |
|     | **Support** | 3.71 | 1.02 |
| A1 | I have happy being in a gifted program in my school. | 4.11 | .91 |
| A2 | Most of my family and friends consider me gifted. | 2.89 | 1.33 |
| A3 | I consider myself to be gifted | 3.91 | .94 |
| A4 | Most of my family and friends are gifted. | 3.97 | .94 |
| A6 | Our schools offer appropriate special education services for the gifted. | 3.65 | 1.00 |
|     | **Acceleration** | 3.56 | 1.10 |
| A17 | Most gifted children who skip a grade have difficulties in their social adjustment to a group of older students. | 3.69 | 1.21 |
| A18 | Children who skip a grade are usually pressured to do so by their parents. | 3.58 | .97 |
| A19 | When skipping a grade, gifted students miss important ideas | 3.93 | 1.13 |

## Predictors of attitudes

Path analysis was computed to ascertain whether self-concepts could contribute to the variance in attitudes (see Fig 2). An initial CFA showed a significant correlation between overall attitudes and self-concepts ($r = 0.81$).

The contribution of self-concept to the variance in attitudes was assessed. First, regarding the attitude subscale of support, the following subscales of self-concept significantly contributed to its variance: scholastic competence ($b = 0.17$, $p = 0.003$), social competence ($b = 0.13$, $p = 0.005$), physical appearance ($b = 0.11$, $p = .001$), behavioral conduct ($b = 0.26$, $p = .001$), and global self-worth ($b = 0.11$, $p = .02$). Second, concerning the attitude subscale of acceleration, only one predictor made a significant contribution to the variance: scholastic competence ($b = −0.20$, $p = .04$).

## Moderation

To comprehend the impact of gender and ability area on the relationship between attitude and well-being, moderation analyses were performed. First, gender was operationalized as a moderator to understand its influence on the relationship between acceleration and self-concept. The initial model looked at how gender affected one subscale of attitude and self-concept. Gender had a direct effect only on physical appearance ($b = .03$, $p = . 04$). However, gender did not influence acceleration ($b = −.24$, $p = .52$). Concerning the standardized indirect effect, the results showed that gender had no interaction effect on the relationship between acceleration and self-concept.

Second, the moderation effect of gender on the relationship between the second subscale of attitude, support, and well-being was assessed. Regarding the standardized direct effect, gender had no direct influence on support ($b = −.10$, $p = .55$). Furthermore, the association between support for gifted education and well-being was unaffected by gender.

The interaction effect of a high-ability area on the link between attitude and well-being was also assessed. For instance, the area of high ability had a standardized direct effect on social competence ($b = .08$, $p = .01$) and scholastic competence ($b = −.60$, $p = .01$). Additionally, the area of high ability had no direct influence on acceleration ($b = −.13$, $p = .61$) and support for gifted education ($b = −.22$, $p = .63$). Areas of high ability did not have a standardized indirect effect on the association between attitude and self-concept. Similar findings were observed regarding the degree to which grade level and age moderated the link between attitudes and self-perceptions.

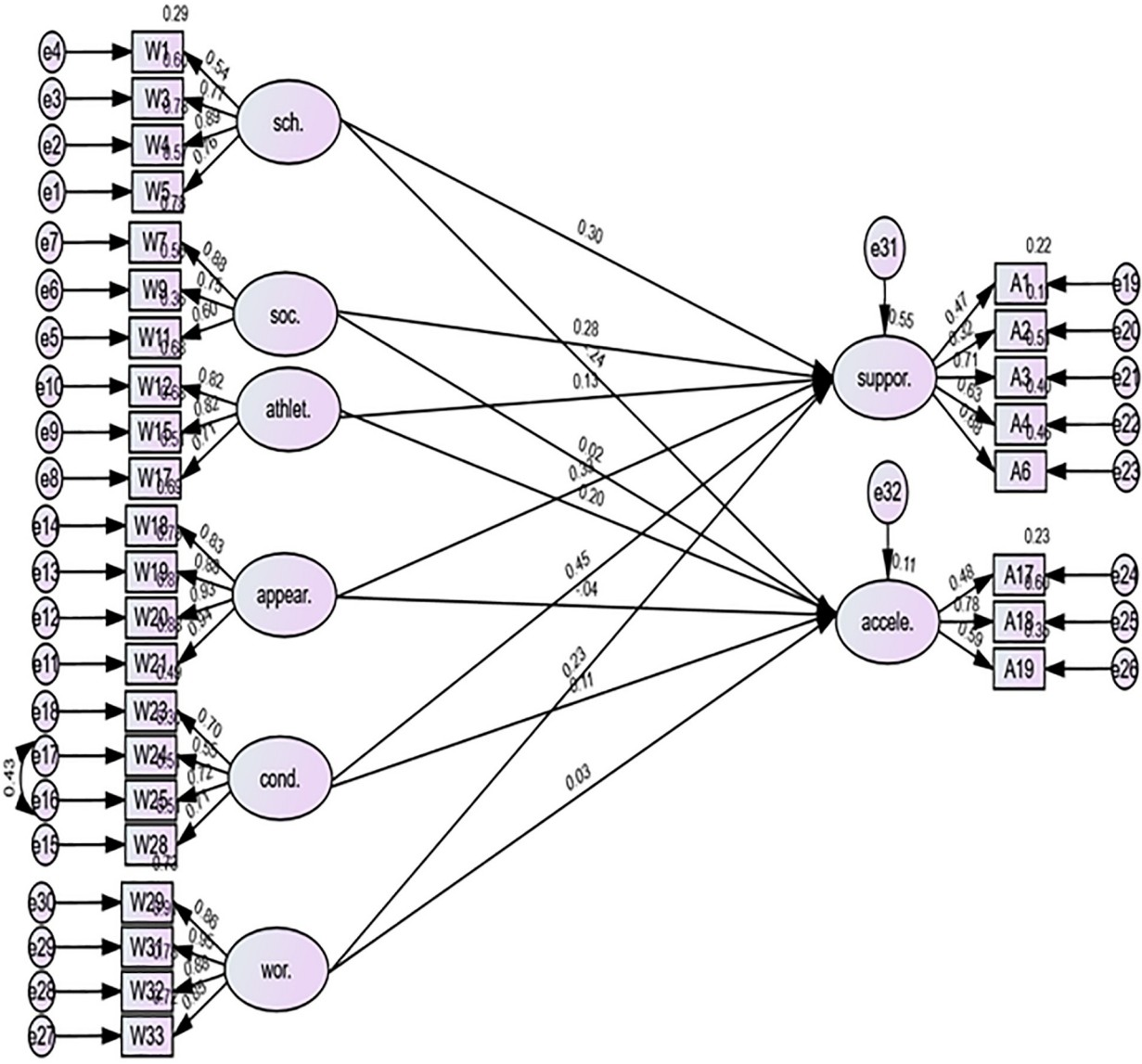

**Fig 2. Contribution of self-conception in the variance in attitude.**

## Discussion

In this study, Ajzen's [20] TPB was adopted to understand the relationship between the attitudes and self-concepts of GT students in a novel cultural context, the UAE. The CFA used in the study, whose factorial validity was supported, was used to test the structural validity of the OGE and SPPC scales. The results showed a six-factor structure for the SPPC and a two-factor structure for the OGE. Although modifications were deployed to eliminate reduced items whose erroneous covariances affected the model fit indices [50], the tools were validated in a context other than the usual Western context, where such studies are usually conducted. The validation process involved the observation of numerous parameters (chi-square, CFI, TLI, modifications, etc.) [5, 48, 50] before ascertaining their validity. The results demonstrate the potential of the SPPC and OGE measures to examine talented students' attitudes and self-perceptions in the UAE and other Arabian countries.

The results of the study supported Hypothesis I and demonstrated a relationship between self-concept and attitude. This finding supports Ajzen's [20] proposition of a relationship between individuals' attitudes and self-concepts in explaining their intention toward a given behavior. The results are consistent with earlier studies on inclusive education, which revealed a favorable association between people's attitudes and perceptions of their own actions [25, 26]. The findings suggest that in the UAE context, as the attitudes of GT students toward participation in gifted programs improve, the students' self-concepts may also improve. The UAE government's top priority is to educate and prepare future generations to support the nation's development [40]. However, achieving this goal relies heavily on GT students, who must be identified and placed in requisite programs where their skills will be nurtured. Unquestionably, if educators make conscious efforts to improve students' attitudes, which might affect GT students' self-perceptions, and vice versa, such a goal could be accomplished.

Although Ajzen [20] argued about the potential for demographic variables to impact intention toward a given behavior, this position was not supported in this study. Background factors were originally thought to have an impact on how GT students in the UAE viewed themselves with respect to their attitudes. However, the findings showed that no demographic variables had an association with or moderated the relationship between attitude and self-concept. This conclusion conflicts with earlier GT studies that found that attitudes and self-perceptions were influenced by demographic factors [6–8]. The study finding is also inconsistent with previous inclusive education research, which reported the influence of demographic variables on attitudes and self-concepts toward learning [25, 26]. In the current study, the participants were from private schools only, which could influence the trend identified in this study. Therefore, future studies could examine the attitudes and self-perceptions of GT students in both public and private schools.

One curious finding was the lack of a relationship between the two attitude subscales, support and acceleration. Attitudes explain the disposition of participants toward being identified and included in gifted programs. Ajzen [20] contended that belief accumulation explains attitudes toward a given behavior, such as being involved in gifted programs. However, in this study, the findings showed no relationship between acceleration and support for gifted programs. Notably, the study's participants engaged in enrichment activities that could have influenced their results. There is a possibility that they were aware of the enrichment program and not the acceleration program. Anecdotal data indicate that many schools in the UAE discourage acceleration, particularly in private institutions. However, the findings indicated that the participants were unsure of support for gifted programs. The UAE government is currently looking at programs to enhance GT students' development [19, 40]. This could trigger or expedite discussions among UAE educators on the best methods and support programs for GT students in the country.

The results appear to support the multidimensionality of self-concept in the context of the current investigation. According to Winne et al. [34], self-concept should be studied with a multiple lens to develop a holistic understanding. For example, the effectiveness of studies may be impacted by both classroom procedures and extracurricular activities. The association between the self-concept subscales seems to provide useful guidance for education practices in the UAE, even if the purpose here is not to establish causation. It is useful for educators in the UAE to observe a multiplicity of factors that could negatively impact the confidence of GT students. Indeed, a significant correlation was observed between self-worth and behavior management. The tendency of GT students to exhibit unfavorable behavior has been mentioned in some literature, which might affect how well they succeed [1, 2]. In the UAE context, to avert this situation, policymakers might consider developing the self-worth of individuals by building their self-confidence and beliefs, which could positively affect their behavioral conduct.

## Conclusion

The study was guided by Ajzen's [20] TPB. The main objective was to investigate the connection between GT students' attitudes and self-perceptions in the UAE context. The TPB [20] makes two important propositions: (a) there is a relationship between the predictors and intentions toward a given behavior, and (b) background variables have an influence on the predictors. In the research on GT students in the UAE, these two hypotheses were put to the test. One proposition was supported by the study's findings, whereas the other was not. The study's findings showed a substantial relationship between GT students' self-concepts and attitudes. Conversely, demographic variables did not influence the relationship between attitudes and self-concepts. In conclusion, the study supports the idea that attitudes and self-concepts are crucial in creating a successful program for GT students in the UAE and other similar settings.

## Limitations and recommendations for future research

It is impossible to generalize our findings because the study was constrained in certain ways. First, the researchers recruited GT students from private schools participating in an enrichment program in one of the seven states in the UAE. Hence, the interpretation of the study's findings is limited to students from private schools participating in an enrichment program. However, the UAE is a diverse society with students from varied backgrounds and cultures enrolled in private schools. Moreover, one regulating agency, the Ministry of Education, oversees and approves teaching standards and maintains schools. Although the findings could reflect the situation more generally in the country, future research could extend this study to public schools to compare the attitudes and self-concepts of GT students enrolled in both private and public schools. Second, the data were collected online, and the opportunity to ask the students questions for clarification may have been limited. Future research could use qualitative techniques to develop thorough insights into the experiences of GT students. Third, the findings are an account of GT students only, and it is uncertain whether non-GT students could share the same level of attitudes and self-concepts. Future research could examine how GT and non-GT students view themselves to develop holistic insights into their self-concepts and attitudes toward gifted education.

Nevertheless, the findings could have implications for policymaking in the UAE and other similar cultural contexts. First, policymakers could consider developing awareness and training programs for GT students geared toward enabling them to understand their contributions to national development. Additionally, GT students could be made more aware of why it is important for them to participate in gifted programs. Furthermore, GT students could be involved in extracurricular activities to help improve their self-concepts. For instance, aside from academic programs, educators could consider organizing athletic programs and related activities for GT students to improve their self-concepts. Third, educators could consider clarifying the scope of gifted programs in the UAE. This could include the criteria for acceleration and fine-tuning acceleration programs, which are conterminous with contextual realities. The UAE is working to enhance students' well-being, and measures such as those suggested above could assist in fostering an environment that is favorable to the development of GT students.

## Acknowledgments

We would like to thank the Abu Dhabi Department Knowledge and Education for their support during recruitment and data processing. Our heartfelt gratitude goes to all students and parents who participated in this study.

## Author Contributions

**Conceptualization:** Maxwell Peprah Opoku, Ashraf Moustafa, Negmeldin Alsheikh, Noora Anwahi, Mariam Aljaberi, Thara Alkhateri, Aysha Almeqbaali, Hala Elhoweris.

**Data curation:** Maxwell Peprah Opoku, Ashraf Moustafa.

**Formal analysis:** Maxwell Peprah Opoku.

**Investigation:** Maxwell Peprah Opoku.

**Methodology:** Maxwell Peprah Opoku, Ashraf Moustafa, Negmeldin Alsheikh, Noora Anwahi, Mariam Aljaberi, Thara Alkhateri, Aysha Almeqbaali, Hala Elhoweris.

**Project administration:** Maxwell Peprah Opoku.

**Resources:** Maxwell Peprah Opoku.

**Writing – original draft:** Maxwell Peprah Opoku, Ashraf Moustafa, Negmeldin Alsheikh, Noora Anwahi, Mariam Aljaberi, Thara Alkhateri, Aysha Almeqbaali, Hala Elhoweris.

**Writing – review & editing:** Maxwell Peprah Opoku, Ashraf Moustafa, Negmeldin Alsheikh, Noora Anwahi, Mariam Aljaberi, Thara Alkhateri, Aysha Almeqbaali, Hala Elhoweris.

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
