## [Decision Letter · Decision Letter 0]

11 Mar 2024

PONE-D-24-06290The Nexus Between Attitudes and Self-Concept of Gifted Students in an Arab ContextPLOS ONE

Dear Dr. Opoku,

Thank you for submitting your manuscript to PLOS ONE. After careful consideration, we feel that it has merit but does not fully meet PLOS ONE’s publication criteria as it currently stands. Therefore, we invite you to submit a revised version of the manuscript that addresses the points raised during the review process.

We look forward to receiving your revised manuscript.

Kind regards,

Sanaa Kaddoura, Ph.D.

Academic Editor

PLOS ONE

Journal Requirements:

   "Office of the Associate Provost for Research at United Arab Emirates University."  

3. For studies involving third-party data, we encourage authors to share any data specific to their analyses that they can legally distribute. PLOS recognizes, however, that authors may be using third-party data they do not have the rights to share. When third-party data cannot be publicly shared, authors must provide all information necessary for interested researchers to apply to gain access to the data. (https://journals.plos.org/plosone/s/data-availability#loc-acceptable-data-access-restrictions) 

Reviewers' comments:

Reviewer's Responses to Questions

**Comments to the Author**

1. Is the manuscript technically sound, and do the data support the conclusions?

Reviewer #1: Partly

Reviewer #2: Yes

2. Has the statistical analysis been performed appropriately and rigorously? 

Reviewer #1: Yes

Reviewer #2: Yes

3. Have the authors made all data underlying the findings in their manuscript fully available?

Reviewer #1: No

Reviewer #2: Yes

4. Is the manuscript presented in an intelligible fashion and written in standard English?

Reviewer #1: No

Reviewer #2: Yes

5. Review Comments to the Author

Reviewer #1: I highly recommend mentioning the selection method related to the sampling and citing it with relevant literature.

*The current study drew on Gifted students from private schools as they were running enrichment program for GT students. The decision to include students in private schools was based on the fact that permission ADEK granted permission to include students on their high-ability roster. *

**I highly recommend mentioning the selection method related to the sampling here and citing it with relevant literature. **

**I highly recommend redrawing Figure One for clarity.**

**All the references are extremely outdated; I highly recommend using recent resources. **

**Here are some recommendations **

Choosing between the theory of planned behavior (TPB) and the technology acceptance model (TAM)
Control Interactions in the Theory of Planned Behavior: Rethinking the Role of Subjective NormTheory of Planned Behavior The Handbook of Behavior Change
**Kindly replace the keywords with these terms and phrases to better suit the article’s focus. **

**For instance, keeping gifted and talented students phrase suits the article however, I would suggest removing ( enrichment; acceleration; United Arab Emirates) since the paper does not tackle all the schools in UAE, so we better not generalize the paper’s findings.**

**Major language edits need to be performed.  Major revisions are needed when it comes to language correctness, kindly either use Grammarly premium or consult external editors to perfect the language to make sure it meets the highest standards.**
I find this section as being irrelevant and serves an academic purpose

  “* The UAE is an Arab country in Western Asia which is made up of seven emirates: Abu Dhabi, Ajman, Dubai, Fujairah, Sharjah, Ras Al Khaimah, and Umm Al Quwain. Since independence in 1971, the leadership of the country has placed education at the heart of national development [19]. The UAE adopted inclusive education practices at the turn of the 21st century. Specifically, the central government provides educational plans for all students [37]. In 2008, the “School for All” policy was launched by the Ministry of Education, encouraging schools to provide suitable teaching services to high-achieving children [19]. *

**Instead, the authors could have mentioned a more detailed overview or a comparative approach to how the educational system witnessed massive changes over time. The references do not provide credibility to the topic, as they are extremely outdated and need to be replaced. **

**The research design is not clear.**

**The methodology section’s structure needs to be revised. **
“ The instrument was made up of three sections. The first section was used to collect demographic”. **Kindly use academic referencing here; it cannot be stated as a passive statement.” **
**Following the standard academic publishing outline, it is advised to follow the following paper structure**

**Introduction**

**Background **

**Literature Review**

**Significance \\ Contribution **

**Methods**

**Design **

**Data Collection Data Analysis**

**Discussion **

**Conclusion**

**Recommendations and Future Research **

**It is more advised to clearly identify these sections as mentioned in  the above subheading divisions. **

**Major sentence structure revisions are needed.**
For instance

Google Forms was employed as a virtual platform for data collection because restrictions brought by COVID-19 pandemic.  (This sentence is confusing).

**A better version would be **

“ Google Forms were employed for data collection because of mobility challenges imposed  by COVID-19 pandemic.”

**Here, another question arises: when was the study conducted? The pandemic was in 2020-2022, and we are now in the 2024-2025 academic year. As it is stated in the paper**

**The data was collected from May 2022 to July 2022. Where most educational institutions went back to Hybrid and onsite attendance which means there is no clear justification for doing online surveys. You may remove the mention of covid 19 in the above statement and relate it to the reasoning behind selecting such a method. **

**Major language confusions need to be clarified. For example, consider the examples below **

*“ Non-Western countries’ contribution of research in gifted education is limited, despite the fact that some have adopted policies to foster the growth of exceptional students.”*

**Based on what did you come up with this claim? Which non-western countries? **

**In the methodology section, these need to be clarified **

150 high school students in one of the seven emirates in the United Arab Emirates to test two hypotheses. Which location? Which school? This cannot be stated as a general statement.

**Criteria bias**

*“The participants of this study were GT high school students nominated by their schools to participate in an enrichment program.”*

**This seems a bit unreliable, for instance, what is the selection method used?  **

**Major language highlights that demand immediate attention**

**Subject-verb agreement**

**Kindly note that in many cases, needless capital letters are used, so please revise. **

Reviewer #2: Th researchers apply ethical consideration. The data are analyzed comprehensively.

The researchers need to edit the first research question as follows:

'What are the differences between..."

- Indent the beginning of all paragraphs.

6. PLOS authors have the option to publish the peer review history of their article (what does this mean?). If published, this will include your full peer review and any attached files.

Reviewer #1: **Yes: **Fatima Al Husseiny

Reviewer #2: **Yes: **Jana Mohammad Saab

---

## [Decision Letter · Decision Letter 1]

23 Apr 2024

PONE-D-24-06290R1The Nexus Between Attitudes and Self-Concept of Gifted Students in an Arab ContextPLOS ONE

Dear Dr. Opoku,

Thank you for submitting your manuscript to PLOS ONE. After careful consideration, we feel that it has merit but does not fully meet PLOS ONE’s publication criteria as it currently stands. Therefore, we invite you to submit a revised version of the manuscript that addresses the points raised during the review process.

We look forward to receiving your revised manuscript.

Kind regards,

Sanaa Kaddoura, Ph.D.

Academic Editor

PLOS ONE

Journal Requirements:

Reviewers' comments:

Reviewer's Responses to Questions

**Comments to the Author**

1. If the authors have adequately addressed your comments raised in a previous round of review and you feel that this manuscript is now acceptable for publication, you may indicate that here to bypass the “Comments to the Author” section, enter your conflict of interest statement in the “Confidential to Editor” section, and submit your "Accept" recommendation.

Reviewer #1: All comments have been addressed

Reviewer #2: All comments have been addressed

2. Is the manuscript technically sound, and do the data support the conclusions?

Reviewer #1: Yes

Reviewer #2: Yes

3. Has the statistical analysis been performed appropriately and rigorously? 

Reviewer #1: Yes

Reviewer #2: Yes

4. Have the authors made all data underlying the findings in their manuscript fully available?

Reviewer #1: No

Reviewer #2: Yes

5. Is the manuscript presented in an intelligible fashion and written in standard English?

Reviewer #1: No

Reviewer #2: Yes

6. Review Comments to the Author

Reviewer #1: I highly recommend you reviewing the language correctness through Grammarly. The paper needs language edits, for this, working on the sentence structure and adjusting the language flaws will enhance the quality of the paper.

Reviewer #2: The authors did the previous edits that were required previously. I do recommend that this paper is ready to be published.

7. PLOS authors have the option to publish the peer review history of their article (what does this mean?). If published, this will include your full peer review and any attached files.

Reviewer #1: **Yes: **Fatima Amine Al Hussseiny

Reviewer #2: **Yes: **Jana Mohammad Saab

---

## [Author Response · Author response to Decision Letter 1]

30 Apr 2024

Dear editor,

Thanks for considering our paper and we appreciate feedback from the reviewers. Below is our point-by-point response to the reviewers’ comments. Thanks once again and looking forward to your feedback. 

Comments 

Response

Thanks once again. We have cross-checked and ensured that all references are cited appropriately. 

1. If the authors have adequately addressed your comments raised in a previous round of review and you feel that this manuscript is now acceptable for publication, you may indicate that here to bypass the “Comments to the Author” section, enter your conflict of interest statement in the “Confidential to Editor” section, and submit your "Accept" recommendation.

Reviewer #1: All comments have been addressed

Reviewer #2: All comments have been addressed

Response: Thanks for your feedback on our submission.

2. Is the manuscript technically sound, and do the data support the conclusions?

Reviewer #1: Yes

Reviewer #2: Yes

Response: Thanks for your feedback on our submission.

3. Has the statistical analysis been performed appropriately and rigorously? 

Reviewer #1: Yes

Reviewer #2: Yes

Response: Thanks for your feedback on our submission.

4. Have the authors made all data underlying the findings in their manuscript fully available?

Reviewer #1: No

Reviewer #2: Yes

Response: We have added a data availability statement (The datasets generated and analysed during the current study are not publicly available due to ethical restrictions but are available from the corresponding author upon reasonable request). 

5. Is the manuscript presented in an intelligible fashion and written in standard English?

Reviewer #1: No

Reviewer #2: Yes

Response to comments: We have attached an invoice showing the editing of the paper by a reputable language editor. We are confident all errors have been correctly. 

6. Review Comments to the Author

Reviewer #1: I highly recommend you reviewing the language correctness through Grammarly. The paper needs language edits, for this, working on the sentence structure and adjusting the language flaws will enhance the quality of the paper.

Reviewer #2: The authors did the previous edits that were required previously. I do recommend that this paper is ready to be published.

Response to comments: We have attached an invoice showing the editing of the paper by a reputable language editor. We are confident all errors have been correctly.

---

## [Decision Letter · Decision Letter 2]

22 May 2024

The Nexus Between Attitudes and Self-Concept of Gifted Students in an Arab Context

PONE-D-24-06290R2

Dear Dr. Opoku,

We’re pleased to inform you that your manuscript has been judged scientifically suitable for publication and will be formally accepted for publication once it meets all outstanding technical requirements.

Kind regards,

Sanaa Kaddoura, Ph.D.

Academic Editor

PLOS ONE

Additional Editor Comments (optional):

Reviewers' comments:

Reviewer's Responses to Questions

**Comments to the Author**

1. If the authors have adequately addressed your comments raised in a previous round of review and you feel that this manuscript is now acceptable for publication, you may indicate that here to bypass the “Comments to the Author” section, enter your conflict of interest statement in the “Confidential to Editor” section, and submit your "Accept" recommendation.

Reviewer #1: All comments have been addressed

2. Is the manuscript technically sound, and do the data support the conclusions?

Reviewer #1: Yes

3. Has the statistical analysis been performed appropriately and rigorously? 

Reviewer #1: Yes

4. Have the authors made all data underlying the findings in their manuscript fully available?

Reviewer #1: No

5. Is the manuscript presented in an intelligible fashion and written in standard English?

Reviewer #1: Yes

6. Review Comments to the Author

Reviewer #1: After reviewing this paper again, I confirm that the authors adjusted previous comments and enhanced the paper's outcome. I only have two concerns:

1- Kindly make the data fully available as per PLOS One guidelines

2- Perform another round of language editing for general minor correctness.

7. PLOS authors have the option to publish the peer review history of their article (what does this mean?). If published, this will include your full peer review and any attached files.

Reviewer #1: **Yes: **Fatima Al Husseiny

---

## [Editor Report · Acceptance letter]

4 Jun 2024

PONE-D-24-06290R2 

PLOS ONE

Dear Dr. Opoku, 

I'm pleased to inform you that your manuscript has been deemed suitable for publication in PLOS ONE. Congratulations! Your manuscript is now being handed over to our production team.

Kind regards, 

on behalf of

Dr. Sanaa Kaddoura 

Academic Editor

PLOS ONE